# A Simple Method Using an Allometric Model to Quantify the Carbon Sequestration Capacity in Vineyards

**DOI:** 10.3390/plants12050997

**Published:** 2023-02-22

**Authors:** Rui Song, Zongwen Zhu, Liang Zhang, Hua Li, Hua Wang

**Affiliations:** 1College of Enology, Northwest A&F University, Xianyang 712100, China; 2China Wine Industry Technology Institute, Yinchuan 750021, China

**Keywords:** carbon storage, distribution features, perennial organs, vineyard ecosystem, winegrape

## Abstract

Winegrapes are an important component of agroecosystems. They are endowed with great potential to sequester and store carbon to slow down greenhouse gas emissions. Herein, the biomass of grapevines was determined, and the carbon storage and distribution features of vineyard ecosystems were correspondingly analyzed using an allometric model of winegrape organs. Then, the carbon sequestration of Cabernet Sauvignon vineyard in the Helan Mountain East Region was quantified. It was found that the total carbon storage of grapevines increased with vine age. The amounts of the total carbon storage in the 5-year-old, 10-year-old, 15-year-old, and 20-year-old vineyards were 50.22 t·ha^−1^, 56.73 t·ha^−1^, 59.10 t·ha^−1^, and 61.06 t·ha^−1^, respectively. The soil held the majority of the carbon storage, which was concentrated in the top and subsurface layers (0–40 cm) of the soil. Moreover, the biomass carbon storage was mainly distributed in the perennial organs (perennial branches and roots). In young vines, carbon sequestration increased each year; however, the increased rate in carbon sequestration decreased with winegrape growth. The results indicated that vineyards have a net carbon sequestration capacity, and within certain years, the age of grapevines was found to be positively correlated with the amount of carbon sequestration. Overall, the present study provided accurate estimations of the biomass carbon storage in grapevines using the allometric model, which may help vineyards become recognized as important carbon sinks. Additionally, this research can also be used as a basis for figuring out the ecological value of vineyards on a regional scale.

## 1. Introduction

Climate change exposes ecosystems to strong and rapid changes in the conditions of their environmental boundaries, mainly due to the increase in greenhouse gas emissions [1]. The concentration of CO_2_ in the air has increased significantly as a result of human activities, fossil fuel use, and land use changes [2,3]. Climate change affects human health [4], alters vegetation phenology [5], reduces biodiversity [6], and increases poverty [7]. Replacing fossil fuels with non-carbon energy can reduce carbon emissions in the atmosphere, which is considered a potential mitigation measure for climate change [8]. Achieving carbon neutrality also involves efforts to mitigate climate change through carbon sequestration [9]. Most studies concerning the terrestrial carbon cycle focus on the forest and grassland ecosystems [10,11,12], while few studies have been conducted on agroecosystems. Indeed, the quantity of carbon sequestered in agroecosystems is critical to the sustainability of regional ecosystems and the global carbon balance [13,14]. Agroecosystems offer two main options for increasing carbon storage: growing crops on the currently uncultivated land, or allowing the existing cropland to accumulate higher biomass [15]. In addition to the economic value of the fruit produced, orchards represent an important facet of the overall agricultural ecosystem, and furthermore provide ecosystem service functions such as soil and water conservation, landscape protection, carbon fixation, and oxygen release [16]. Additionally, they are easier to manage than non-commercial forest and grassland ecosystems. A number of interventional strategies, such as lowering fertilizer use, controlling yield, mulching cultivation, or encouraging service crop planting, can be adopted to increase soil fertility and productivity, while reducing carbon emissions [17,18].

Winegrapes (*Vitis vinifera* L.) are one of the cash crops being developed rapidly worldwide, and they play an important role in the agricultural economy [19]. In 2021, the total area of global grape cultivation was 7.85 × 10^6^ ha, of which 7.83 × 10^5^ ha was cultivated in China, making China third largest grape-growing country in the world [20]. More attention has been paid to the sustainable development of the global wine industry, which highlights the growing interest of both customers and the industry as a whole, in reducing or offsetting the greenhouse gas emissions caused by grapewine production [21,22]. Thus, a growing number of studies in recent years have explored carbon storage within vineyard ecosystems [23,24]. Carbon storage refers to the amount of carbon stored in each carbon pool of an ecosystem [25]. Most research on carbon storage in vineyards has focused on the soil [26,27], which is the most important component of the vineyard ecosystem. Meanwhile, irrigation and fertilization management, shaping and pruning techniques, planting density, inter-row grass, intra-row branch coverage, and tree age all have a great influence on the carbon absorption and emission characteristics of the ecosystem [22,28]. Considerable efforts have also been made to examine above-ground carbon fixation at the vineyard, with efforts to estimate the carbon storage of grapevines [29,30]. A trustworthy method for estimating carbon storage, from individual vines to the entire farm scales, may be provided by allometric correlations between trunk diameter and vine biomass [11,31,32]. For example, Morandé et al. successfully studied the carbon storage of 15-year-old Cabernet Sauvignon vineyards using an allometric model, and found that grapevines achieved an average carbon storage level of 12.3 t·ha^−1^ [33]. These findings indicate that well-cultivated vineyards may operate as carbon sinks, through organic carbon sequestration. Due to their unique structural characteristics, vines may be allowed to store more organic carbon than annual crops [34].

Some efforts have also been made to estimate the annual carbon sequestration in different agroecosystems [8,35,36]. Sierra et al. presented a formal definition of carbon sequestration as the integral of an amount of carbon removed from the atmosphere stored over the time horizon that it remains within an ecosystem [37]. This provides a more intuitive expression of the role of agroecosystems in controlling global greenhouse gas emissions. Recent studies have investigated the carbon content of grape biomass, using destructive methods to understand plant carbon allocation and carbon storage capacity [29,38]. These methods, however, disregard carbon sequestration as a dynamic process, and only provide discrete information on the vineyard state at any given time. Instead, annual carbon sequestration in vineyards is crucial for assessing the role of viticulture in the global carbon budget, making the comprehensive assessment of vineyard carbon storage and annual carbon sequestration beneficial for more accurate carbon accounting of the wine industry.

Herein, the study was carried out to estimate the biomass of the above-ground grapevines, and assess the potential of grapevines for carbon sequestration using an allometric model. The carbon storage rates in vines were estimated based on the annual growth increments in carbon sequestration in four vineyards of different ages; how carbon storage varied according to the vine age was also examined. The initial hypothesis was that the quantification of carbon storage in vineyard systems reflects the viability of perennial crops, which positively offsets carbon emissions associated with grapewine production. Hence, to test this hypothesis, our study was designed with three primary objectives: (1) to provide a quantitative understanding of the carbon storage and distribution characteristics of vineyards of different ages in China; (2) to define allometric relationships that enable growers and land managers to quickly assess vineyard carbon stocks; and (3) to examine the effect of changing grapevine age on carbon sequestration capacity. The analysis further indicated that a vineyard can function as a carbon sink in the medium-to-long term. The carbon fixed by grapevines could help offset carbon emissions associated with winegrape production, which should be factored into agricultural and environmental policies. This research assists in promoting the development of the Chinese wine industry, and provides a theoretical reference for the coordination and sustainable development of the economy, society, and environment.

## 2. Materials and Methods

### 2.1. Study Site

The present study was conducted in the Qingtongxia (YaDai Winery) and Yinchuan (Horticultural Institute) winegrape planting areas in the Helan Mountain East Region of Northwest China in September 2021 (105°43′45″–106°42′50″ E, 37°28′08″–37°37′23″ N) (Figure 1). The average frost-free period is 199 days, the average annual precipitation is 175.9 mm, the average annual evaporation is 1864.5 mm, the average annual temperature is 9.2 °C, the average annual daily temperature difference is 13.7 °C, and the annual sunshine duration is 2900–3550 h. The texture of the soil in this region is a sandy loam texture (45% clay, 30% silt, and 25% sand) [39].

We selected four vineyards of different ages, all of which are Eurasian species (*Vitis vinifera*), namely ‘Cabernet Sauvignon’ varieties. The 5-year-old vineyard was planted in April 2016, and the 10-year-old, 15-year-old, and 20-year-old vineyards were planted in April 2011, 2006, and 2001, respectively. All of these vineyards were taken care of in a consistent way, with the plants arranged in north–south rows with a row spacing of 3 m and a vine spacing of 1.0 m, using a single-cane “Dulonggan” cultivation approach with an average planting density of 3335 vines·ha^−1^. Totals of 38.0 kg·ha^−1^ of nitrogen fertilizer, 30.0 kg·ha^−1^ of phosphate fertilizer, 8.0 kg·ha^−1^ of potassium fertilizer, and 16.0 kg·ha^−1^ of organic manure were used. The annual irrigation water volume is 3900 m^3^·ha^−1^.

### 2.2. Vine and Soil Sample Collection

In September 2021, after the grapes were finished ripening, four typical representative plots of 667 m^2^ of different ages (5-year-old, 10-year-old, 15-year-old, and 20-year-old) were selected from the vineyards. Since the economic value of the vineyards is mainly based on non-wood use, this study was a destructive experiment, and the sample number was, therefore, reduced as much as possible on the basis of meeting the research needs [38]. The base diameters of the vines ranged from 1.5 to 8.0 cm, across the four vineyards surveyed. The trunk diameters between the maximum and minimum values were used to select standard wood of different sizes. For example, the 5-year-old vines were grouped into diameter classes ranging from 1.5 to 5.0 cm, while the 10-year-old, 15-year-old, and 20-year-old vines were grouped into diameter classes ranging from 2.5 to 6.0 cm, 3.2 to 7.8 cm, and 3.5 to 8.0 cm, respectively. Five vines were randomly selected within the corresponding base diameter range in each plot. The whole root excavation method was used to collect the samples [38] (Figure 2). In total, 20 grapevines of different diameters were collected. All the samples were divided into the leave, fruit, stem, cane, perennial branch, and root. After decomposition, the fresh weight of various organs was accurately measured using an electronic scale, and all the samples were then dried at 80 °C until they reached a constant weight, in order to determine the weight and the calculated dry-fresh ratio of the different organs. Subsequently, the samples were crushed with a high-speed grinder for organic carbon determination.

The soil samples were collected around the rhizosphere of grapevines. In order to collect as much root as possible, the excavation diameter was controlled as 1 m, with a depth of 1 m. Four different soil layers, i.e., the 0–20 cm one, the 20–40 cm one, the 40–60 cm one, and the 60–100 cm one, were collected (Figure 2b). Five replicates were collected per layer, and were then combined into a single composite sample. Finally, a total of 80 soil samples were collected. A portion of soil samples were dried in an oven at 105 °C until they reached a constant weight, to determine soil bulk density; the remaining soil samples were sieved at 2.0 mm and dried in an oven at 80 °C until they attained a constant weight, to assess the soil organic carbon content. The organic carbon content of the vines and soil was determined using the Walkley and Black method [40].

### 2.3. Determination and Calculation Method

#### 2.3.1. Selection of the Biomass Model

The allometric growth model depicts the proportionate and coordinated development of the different components within a particular ecosystem; it uses a power function relationship [41]. Considering measurable factors such as the base diameter as independent variables, this very model features a simple structure, a stable parameter estimation value, and a strong predictive ability [42,43]. To this end, the allometric model was hereby selected as the biomass model for the grapevine, which could be calculated using Equation (1):Y = aX^b^
(1)
where X denotes the diameter of the vine trunk (cm); Y, the total biomass of winegrape (kg); and a and b are constants derived from the regression analysis of each organ of the standard vine.

#### 2.3.2. Base Diameter Measurement

In order to truly reflect the size of the trunk diameter of the vineyards of different ages, a total of 500 grapevines of different sizes were randomly investigated, and the diameter of the trunk at 3 cm above ground was measured using a measuring tape, which was denoted as the trunk base diameter. After recording this information, the base diameter was divided into five equal base diameter ranges according to its size, and the number of trees in each base diameter range was counted, in order to estimate the total number of vines per hectare in each base diameter range.

#### 2.3.3. Total Carbon Storage Calculation

The total carbon storage in the vineyard was hereby defined as the sum of carbon storage in a grapevine’s biomass and soil. The allometric model incorporated the trunk base diameter to calculate the various organ biomasses of individual vines, thus enabling the calculation of the total biomass of each organ per unit area, based on the planting density (3335 vines·ha^−1^). The biomass carbon storage of each diameter was defined as the total biomass of each diameter multiplied by the carbon content of all the organs. The sum of the biomass carbon storage of all the diameters was used to determine the carbon storage of the grapevines per unit area.

The soil carbon storage was defined as the total amount of organic carbon present in the soil, which was mainly concentrated within 1 m of the soil [44]. It could be calculated using Equation (2):(2)$$S_{d}=\sum_{i=1}^{d}D_{i}C_{i}H_{i}\;$$
where S_d_ represents the soil carbon storage per unit area within a soil layer of depth d; D_i_ represents the bulk density of the i-th soil layer; C_i_ represents the carbon content of the i-th soil layer; and H_i_ represents the depth of the i-th soil layer.

The soil bulk density could be calculated using Equation (3):BD = G × 100/V × (100 + W) (3)
where BD is the soil bulk density (t·m^−3^); G is the ring knife soil weight; V is the ring tool volume; and W is the sample water content.

#### 2.3.4. Calculation of Carbon Sequestration

The carbon pool difference method [45] was used to calculate the annual carbon sequestration of vineyards, which was based on the difference in the total carbon storage of vineyards of different ages divided by the year difference. It can be calculated using Equation (4):C_S_ = S_a_ − S_b_/a − b (4)
where C_S_ is the carbon sequestration (t·ha^−1^·a^−1^); S_a_ represents the carbon storage of the a-year; S_b_ represents the carbon storage of the b-year; a, b are the corresponding years.

### 2.4. Data Analysis

The experimental data were organized in Microsoft Office Excel 2017; all data were analyzed using IBM SPSS Statistics 21; and the graphs were built using GraphPad Prism 8. For each vineyard of the different ages studied, the monitoring of different parameters (soil bulk density, biomass of various organs, and carbon content of grapevine and soil) was carried out with five repetitions. Shapiro–Wilk and Levene’s tests were used to assess the normality and homogeneity of variance for data distributions, respectively. An association between the biomass and trunk basal diameter of the grapevine was developed using power function regression analysis, and significant differences among the carbon content of the grapevine organs and different layers of soil were tested using one-way ANOVA, followed by least significant difference (LSD) multiple comparisons. A value of *p* < 0.05 was considered significantly different, and a highly significant difference was identified with *p* < 0.01. The results are presented as the mean ± standard deviation.

## 3. Results

### 3.1. Establishment of an Allometric Model of Winegrape Biomass

The fitting degree between the diameter of the trunk and the biomass of each organ was superior (*p* < 0.05), while the coefficients of determination were all greater than 0.8077, and increased with an increase in the grapevine age (Table 1). The biomass of the 10-year-old grapevine had a highly significant fitting relationship with the fruits, canes, and perennial branches (*p* < 0.01), while that of the 15-year-old grapevine had a significant fitting relationship with the leaves, fruits, annual and perennial branches, and roots; that of the 20-year-old grapevine had a significant fitting relationship with the leaves, fruits, annual and perennial branches, and roots. Therefore, using the trunk basal diameter as a measurement index allows for an accurate analysis of the biomass of each organ in the grapevine.

### 3.2. Association between the Grapevine’s Biomass and Trunk Basal Diameter

By establishing the allometric model between the biomass and the trunk basal diameter of the grapevines, it was found that the total biomass at different ages had a superior fitting relationship with the trunk basal diameter, and that the total biomass increased with an increase in basal diameter (Figure 3). The allometric equation of the 5-year-old grapevine was Y = 0.5569x^0.8697^, the R^2^ = 0.8655*, the base diameter of the vineyard trunk was 1.5–4.5 cm, and the biomass of individual vine was 1.0–2.5 kg (Figure 3a). The allometric equation of the 15-year-old grapevine was Y = 0.0952x^2.0175^, the R^2^ = 0.9688**, and the biomass of per plant was 1.5–5.2 kg (Figure 3c). Additionally, it was also found that with an increase in vine age, the fitting degree between the total biomass and the trunk basal diameter improved.

### 3.3. Number of Grapevines of Different Ages in Different Base Diameter Ranges per Hectare

The total number of vines per hectare in each base diameter range was estimated (Figure 4), and the normality of the distribution was tested using the Shapiro–Wilk method. All *p* values were greater than 0.05, which indicated a normal distribution of the diameter classes of vines with different ages. The trunk basal diameters of the 5-year-old grapevines ranged from 1.5 to 4.5 cm, while those of 1186 vines were between 3.6 and 4.2 cm, accounting for 35.56% of the total. In the 10-year-old vineyard, 32.32% of the grapevines had a trunk basal diameter of 3.9–4.5 cm, with an average of 4.11 cm. The trunk basal diameter between 4.6 cm and 6.0 cm accounted for 36.01 %. Nearly one-third of 15-year-old grapevines had a basal diameter between 5.9 and 6.7 cm, and the maximum basal diameter was 7.8 cm. Moreover, more than 78.2 % of the 20-year-old grapevines had a basal diameter greater than 5.3 cm. The distribution histogram intuitively depicted the main distribution range of grapevine trunk diameters of different ages, and provides a basis for future researchers to select the most representative grapevine trunk diameters in the sampling stage, in order to accurately evaluate the total carbon storage of grapevines.

### 3.4. The Biomass of Various Organs in Individual Vines of Different Ages

The biomass of each organ (leaves, fruits, fruit stems, canes, perennial branches, and roots) of the four vineyards of different ages was calculated (Figure 5). It was found that with an increase in the trunk basal diameter, there were differences in the biomass of each organ, and the roots and perennial branches had much more biomass than the other organs, indicating that roots and perennial branches were the major contributors to the biomass of grapevines. The biomass of the roots of the 20-year-old vine was 2.53 times that of the 5-year-old vine. The biomasses of each organ in the 10-year-old and 15-year-old vines were not significantly different, but the total biomass tended to increase.

### 3.5. The Amounts and Distribution Characteristics of Biomass Carbon Storage in Grapevines of Different Ages

The carbon storage of the roots was found to play the most important role in total carbon storage, which was 0.95 t·ha^−1^ for the 5-year-old vineyard, accounting for 43.05% of the total biomass carbon storage. The carbon storage of the perennial branches was 0.54 t·ha^−1^, accounting for 24.35% (Table 2). No matter what age the vineyard was, the carbon storage of the roots and perennial branches accounted for more than 60% of the total biomass carbon storage, indicating the role of the perennial organs (roots and perennial branches) as the main contributors of biomass carbon storage in the vineyard ecosystem. The biomass carbon storage of the grapevines increased with vine age. The biomass carbon storage of the 10-year-old grapevine increased by 1.78 times to that of the 5-year-old grapevine, while that of the 15-year-old grapevine increased by 1.12 times to that of the 10-year-old grapevine.

### 3.6. Amount and Distribution Characteristics of Soil Carbon Storage in Vineyards of Different Ages

The soil carbon content and bulk density at four different soil depths for the vineyards of different ages were measured. The 0–20 cm soil layer had the highest soil organic carbon content and smallest bulk density. With an increase in soil depth, the soil carbon content decreased, and the soil bulk density increased (Table 3). The soil carbon storage was mainly concentrated on the soil surface. As such, the soil carbon storage amounts in the 0–20 cm and 20–40 cm layers of the 10-year-old and 15-year-old vineyards were significantly higher than those in the 40–60 cm and 60–100 cm layers.

### 3.7. Differences in Total Carbon Storage in Vineyards of Different Ages

The total carbon storage in the Cabernet Sauvignon vineyard showed that vineyard carbon storage increased with the ages of the grapevines (Figure 6). In the vineyard ecosystem, the soil carbon pool was the most important contributor to the total carbon storage. The soil carbon storage amounts of the 5-year-old, 10-year-old, 15-year-old, and 20-year-old vineyards were 48.01 ± 4.50 t·ha^−1^, 52.79 ± 1.63 t·ha^−1^, 54.68 ± 1.81 t·ha^−1^, and 55.57 ± 1.90 t·ha^−1^, respectively. The total carbon storage of the 5-year-old vineyard ecosystem was 50.22 t·ha^−1^, while the total carbon storage amounts of the 10-year-old, 15-year-old, and 20-year-old vineyards were 56.73 t·ha^−1^, 59.10 t·ha^−1^, and 61.06 t·ha^−1^, respectively. Compared with the 5-year-old vineyard, the total carbon storage of the 10-year-old, 15-year-old, and 20-year-old vineyards increased by 12.96%, 17.68%, and 21.59%, respectively. All of these data indicated the significant impact of vine age on the carbon storage of a vineyard ecosystem.

### 3.8. Carbon Sequestration of the Vineyard Ecosystem

The research studied the carbon storage of four vineyards of different ages, and used the difference value method to calculate carbon sequestration, which is an important indicator for characterizing the carbon sink capacity of the vineyards [45]. It was found that the carbon sequestration of the vineyards increased with vine age (Table 4). The younger the vine was, the greater the annual increase in carbon sequestration would be. However, the growth rate declined as the vines aged. During the 5 to 10 years of grapevine growth, the carbon sequestration was 1.31 t·ha^−1^·a^−1^, which was 0.28 t·ha^−1^·a^−1^ during the 10 to 15 years of the grapevine growth, and 0.39 t·ha^−1^·a^−1^ during the 15 to 20 years.

## 4. Discussion

Herein, a biomass allometric model was established to quantify the spatial distribution of carbon storage in the vineyards (Table 1). In several investigations of woody plants, comparable allometric correlations were recorded that mostly relied on the trunk diameter or the cross-sectional area to predict the leaf area or the tree biomass [46,47,48]. This study provides a finer and more precise evaluation of grapevine biomass distributions than previous ones. In the case of evaluating the biomass characteristics of vineyards, the vines were divided into multiple diameter categories for modeling. Additionally, the trunk base diameters of 500 grapevines in four plots of different ages were randomly investigated (Figure 4), in order to provide a basis for future researchers to select the most representative grapevine trunk diameters in the sampling stage, and also to provide a reliable reference for reducing destructive sampling. A significant fitting degree between the biomass and trunk basal diameter of each organ of the grapevine was observed, indicating that the biomass of each organ could be accurately analyzed via easily measurable factors. The results show the importance of below-ground biomass for carbon storage, since root biomass makes up 20–40% of the standing biomass, and 40–60% of that in permanent structures (Table 3); this is consistent with the findings of other studies [38,48]. The biomass carbon storage capacities of Cabernet Sauvignon grapevines were as follows: 2.21 t·ha^−1^, 3.94 t·ha^−1^, 4.42 t·ha^−1^, and 5.49 t·ha^−1^ for the 5-year-old, 10-year-old, 15-year-old, and 20-year-old grapevines, respectively (Table 2); these findings are similar to those of Morandé et al., who studied the carbon storage of vineyards in northern California and presented a carbon storage on the vineyard land of 3.0 t·ha^−1^ [33].

Soil carbon storage in vineyards was discovered to be 3–10 times greater than grapevine biomass carbon storage, indicating that soil is the primary component of total carbon storage within vineyard ecosystems. Furthermore, it was also found that soil carbon storage in vineyards was primarily distributed in the top layer of soil (Table 3). Soil carbon storage in the 0–20 cm layer accounted for more than 1/4 of the total soil carbon storage. These results are consistent with the findings of other studies. For instance, Deurer et al. studied the soil carbon stores of a 15-year-old vineyard in the Marlborough region of New Zealand, and found that they were approximately 12 ± 5 t·ha^−1^ at a depth of 0–0.15 m [49]. The organic carbon content of the surface soil was higher than that of other soil layers, and this decreased with soil depth [50]. In the present study, the soil carbon storage amounts of the 5-year-old, 10-year-old, 15-year-old, and 20-year-old vineyards at 0–100 cm were 48.01 ± 4.50 t·ha^−1^, 52.79 ± 1.63 t·ha^−1^, 54.68 ± 1.81 t·ha^−1^, and 55.57 ± 1.90 t·ha^−1^, respectively. Furthermore, Williams et al. found that the soil carbon storage of vineyards in northern California was 84.1 t·hm^−2^ [45], which is higher than that in the present study. This may be attributed to the fact that the soil of the vineyard was subjected to compaction due to the frequent passing of heavy tractors, thus leading to the low organic carbon content of the soil.

The total carbon storage of a vineyard is the sum of the biomass carbon storage, soil carbon storage, and ground litter carbon storage. The carbon storage of the ground cover litter was not considered here; hence, the present study may have underestimated the carbon storage of the vineyard ecosystem. Moreover, the results of the carbon storage in the vineyards of different ages indicated that the amounts of carbon storage increased with vine age. The total carbon storage amounts of the 5-year-old, 10-year-old, 15-year-old, and 20-year-old vineyards were 50.22 t·ha^−1^, 56.73 t·ha^−1^, 59.10 t·ha^−1^, and 61.06 t·ha^−1^, respectively, figures which were slightly different from the results of Chiarawipa et al., who studied the carbon storage of three vineyards of different ages in China [29]. The vineyards they chose were planted with hybrid vine cultivars. Each vine was trained with a Y-shaped trellis system, under a planting density of approximately 4000 vine·ha^−1^. These were all reasons why their results differed from those of our study; therefore, the results of the present study were relatively smaller. In addition, soil carbon storage accounted for the vast majority of the carbon storage in the vineyard, suggesting that soil carbon storage was the most important component in the vineyard ecosystem. The biomass carbon storage was smaller than that in the study conducted by Zhang et al. [38], which was mainly caused the low density of viticulture in this study, as the vineyard adopted a single-cane “Dulonggan” cultivation approach to facilitate the mechanized operation. The results inspire the researchers, on the basis of ensuring the quality of grapevines, and with the requirements of intelligent mechanization operations considered, to pay more attention to the ecological benefits of vineyards, as well as to increasing in planting density and landscape configuration of vineyards, in order to promote the sustainable and high-quality development of the wine industry.

During the growth of grapevines, the carbon sequestration capacities were 1.31 t·ha^−1^·a^−1^ for the 5–10-year-old vineyards, 0.28 t·ha^−1^·a^−1^ for the 10–15-year-old vineyards, and 0.39 t·ha^−1^·a^−1^ for the 15–20-year-old vineyards. Carbon sequestration was strongly correlated with vine age, where the younger the vine was, the greater the relative increase in annual carbon sequestration would be, a finding which was consistent with that of Williams [45]. This result underlines the important role of vineyard systems in mitigating greenhouse gas emissions. Tezza et al. used the eddy correlation method to study the carbon sequestration of a vineyard ecosystem in northeastern Italy from 2015 to 2016, and found that the vineyard ecosystem was a net carbon sink, and that the amount of carbon sequestration was 2.33 t·ha^−1^·a^−1^ [51]. Differences in the approach may have caused his research results to be higher than those of the present study, as they determined the CO_2_ fluxes between vegetation and atmosphere, following the net ecosystem production methodology, instead of assessing annual growth increments in carbon sequestration. These differences could also be attributed to the greater planting density of their vineyards. Additionally, it is suggested that longer time horizons should be considered for future incremental analyses, as this will improve the precision of the vineyard carbon sequestration estimates.

This research employed an allometric model to examine the carbon sequestration and carbon distribution characteristics of vineyards of different ages in semi-arid areas, and the obtained results may become increasingly important for viticulture management and decision-making efforts. Carbon sequestration can vary depending on ambient and climatic conditions [52]; this is mainly because of the differences in photosynthesis and water utilization rates in different climate zones [53], which lead to large differences in the biomass carbon storage of grapevines. Additionally, differences in soil organic matter are also the main factor to be considered in the case of exploring grapevine carbon storage in different climate zones. If this method is widely used in grape-growing systems in different climatic zones, the relevant measurement accuracy will continuously improve, allowing for an easier understanding of spatio-temporal patterns of carbon distribution within vineyards.

## 5. Conclusions

As a special type of vegetation, economic crops feature the characteristics of a shorter investment cycle and a quicker effect when compared to general ecological forests. The vineyard is considered to be a substantial stationary carbon source that can help remove CO_2_ from emissions through carbon storage in vines, as well as in the soil. Herein, an allometric model was established using different standard base diameters, which constituted a powerful tool to determine the biomass in a non-destructive and cost-effective manner using simple vine measurements, making it suitable for quantifying carbon storage and allocation patterns. Additionally, carbon sequestration in the vineyards was studied by continuously calculating the carbon stock differences at equal age intervals. Overall, the results suggest that vineyards are an effective carbon storage source, and show that soil carbon storage is the main component of total carbon storage in the vineyard ecosystem. Furthermore, this indicates that vineyard management practices, in particular the soil fertilization management of the vineyard, may affect the carbon sink function of the vineyard. The annual growth increments in carbon sequestration in vineyards ranged from 0.28 to 1.31 t·ha^−1^·a^−1^, with a younger vine age indicating a greater annual increase in carbon sequestration. The study of the interannual variations in growth rates clarifies the ecological service functions of vineyards. The present study has important implications for assessing the contribution of grapevine culture to CO_2_ balance. A long-term study on carbon sequestration in vineyards in various regions is critical for assessing the contribution of viticulture to the global carbon budget, and to improvements in management strategies and environmental sustainability. The present research did not involve the differences arising from geography, varietals, and management actions (e.g., irrigation regime, cover crop selection, fertilizer). It is expected that future studies will further qualitatively and quantitatively demonstrate how the positive effects of various vineyard management practices increase carbon storage, such as native vegetation conservation, and can contribute to the provision of ecosystem services and other positive environmental benefits.

## Figures and Tables

**Figure 1 plants-12-00997-f001:**
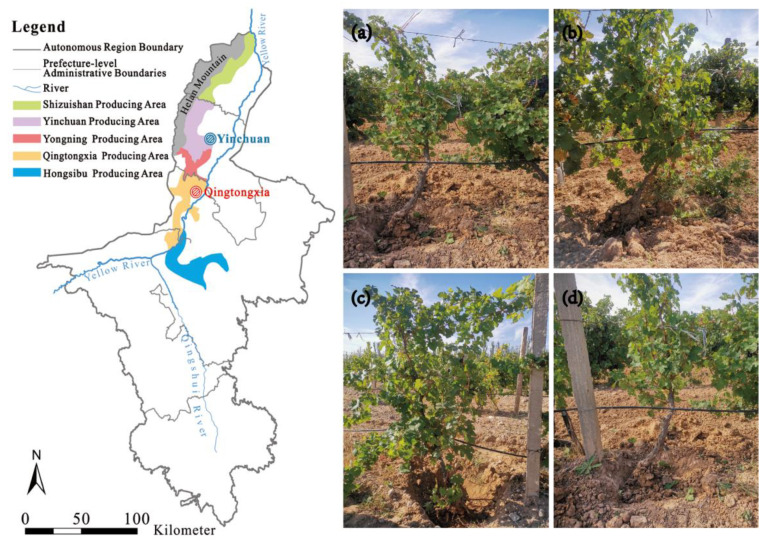
Location of the study site and the four vineyards of different ages within the study area. (**a**) The 5-year-old vineyard; (**b**) the 10-year-old vineyard; (**c**) the 15-year-old vineyard; and (**d**) the 20-year-old vineyard. The 5-year-old samples were collected from the Horticultural Institute at the Yinchuan producing area (the location is marked blue on the map), while the other samples were taken from the YaDai Winery at the Qingtongxia producing area (the location is marked red on the map).

**Figure 2 plants-12-00997-f002:**
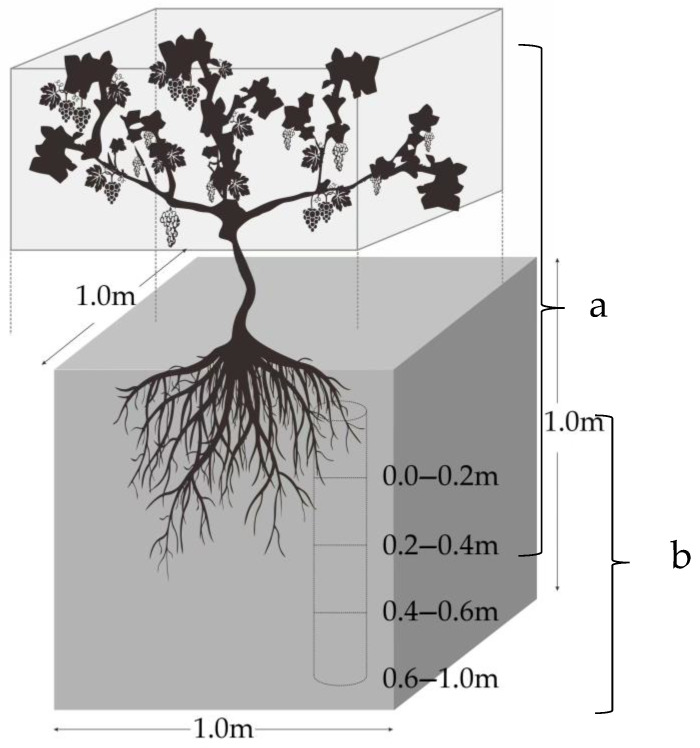
Vineyard sampling diagram. (**a**) Vine sample collection, and (**b**) soil sample collection.

**Figure 3 plants-12-00997-f003:**
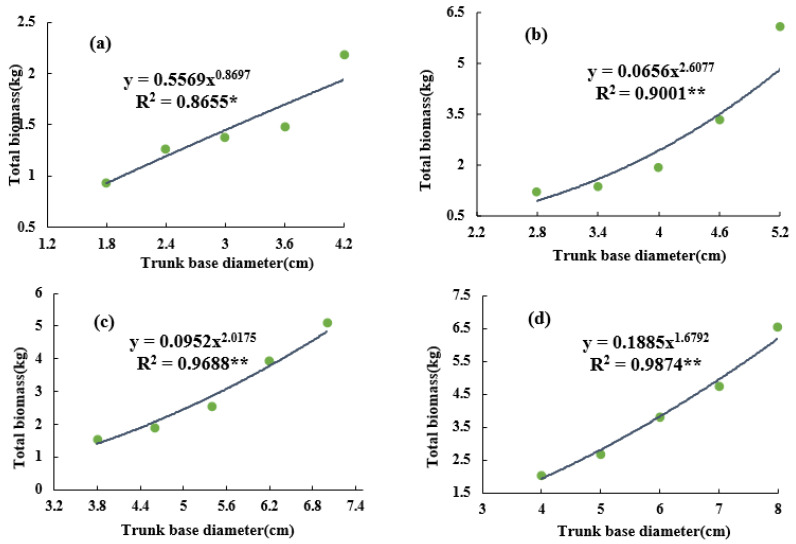
Changes in vine’s biomass of different trunk base diameter classes. (**a**) The 5-year-old vine; (**b**) 10-year-old vine; (**c**) 15-year-old vine; and (**d**) 20-year-old vine. * *p* < 0.05, ** *p* < 0.01. Green circles represent samples with different trunk base diameters.

**Figure 4 plants-12-00997-f004:**
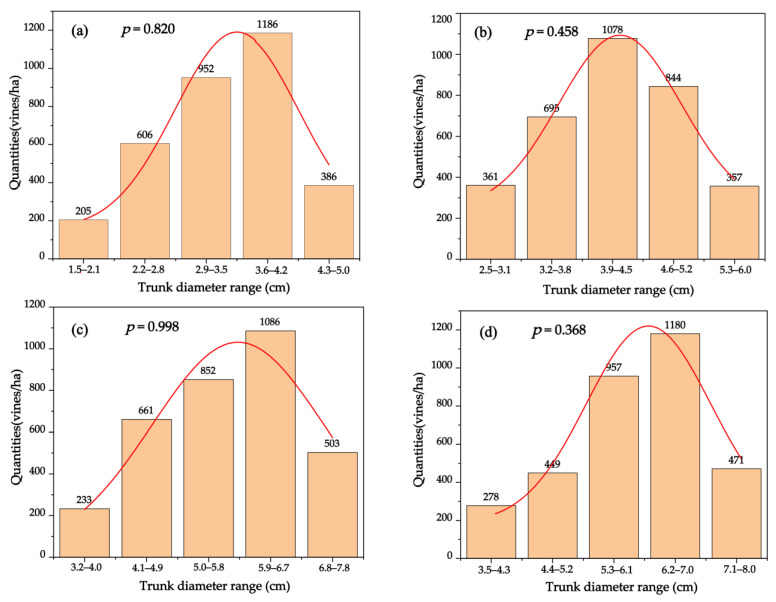
Distribution histogram of the quantities of winegrapes with different ages per hectare in different base diameter ranges. (**a**) The 5-year-old vineyard; (**b**) 10-year-old vineyard; (**c**) 15-year-old vineyard; and (**d**) 20-year-old vineyard. The *p*-values represent the probability that the sample data come from a population conforming to a normal distribution. The *p*-values of the data are all greater than 0.05, suggesting that the data obey a normal distribution, and have homogeneity of variance.

**Figure 5 plants-12-00997-f005:**
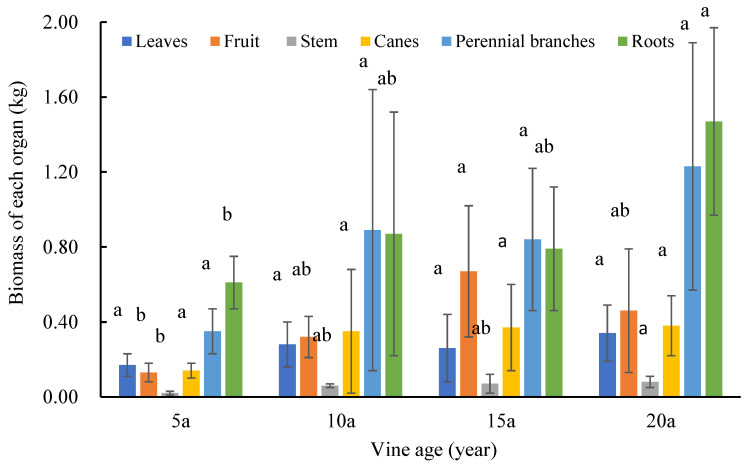
The biomass of various organs in individual vines of different ages. 5a means 5-year-old, 10a means 10-year-old, 15a means 15-year-old, 20a means 20-year-old. Different letters above the columns indicate significant differences between samples in multiple comparisons (*p* < 0.05). The error bars represent the standard deviation of the biological repetition.

**Figure 6 plants-12-00997-f006:**
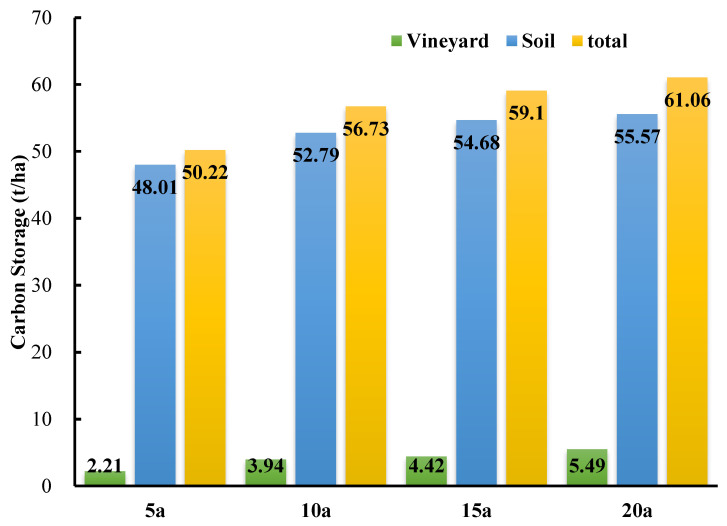
The differences in total carbon storage in vineyard ecosystems of different ages. 5a means 5-year-old, 10a means 10-year-old, 15a means 15-year-old, and 20a means 20-year-old vineyard.

**Table 1 plants-12-00997-t001:** Allometric model-based biomass of grapevine organs.

Vine Age	Allometrics	Leaves	Fruit	Stem	Canes	Perennial Branches	Roots
5-year-old	Model	y = 0.0481x^1.1361^	y = 0.0459x^0.9478^	y = 0.0058x^1.1655^	y = 0.0537x^0.8719^	y = 0.1267x^0.9286^	y = 0.275x^0.763^
	Determination coefficient	0.8258 *	0.9091 *	0.8772 *	0.9002 *	0.907 *	0.8077 *
10-year-old	Model	y = 0.0080x^2.4902^	y = 0.0493x^1.343^	y = 0.0216x^0.792^	y = 0.0018x^3.6312^	y = 0.0079x^3.2602^	y = 0.0290x^2.3563^
	Determination coefficient	0.9113 *	0.9406 **	0.9728 **	0.9360 **	0.9313 **	0.8077 *
15-year-old	Model	y = 0.0018x^2.9243^	y = 0.0208x^2.0449^	y = 0.0004x^2.9118^	y = 0.0073x^2.2897^	y = 0.0851x^1.3100^	y = 0.0436x^1.6383^
	Determination coefficient	0.9127 **	0.9587 **	0.9019 *	0.9176 **	0.9122 **	0.9692 **
20-year-old	Model	y = 0.0190x^1.5904^	y = 0.0076x^2.2211^	y = 0.0044x^1.5712^	y = 0.0250x^1.5015^	y = 0.0223x^2.2023^	y = 0.1681x^1.2044^
	Determination coefficient	0.9677 **	0.8921 *	0.9782 **	0.9279 **	0.9695 **	0.9386 **

Notes: * *p* < 0.05, ** *p* < 0.01. x is the diameter of the vine trunk (cm), y is the biomass of each organ of winegrape (kg).

**Table 2 plants-12-00997-t002:** The biomass carbon storage in grapevines of different ages.

Vine-Age	Organs	Total Biomass of Each Organ (kg·ha^−1^)	Carbon Content (g/kg)	Carbon Storage (t·ha^−1^)	Percentage of Total (%)	Total Carbon Storage (t·ha^−1^)
5-year-old	Leaves	623.40	432.38 ± 4.64 b	0.27	12.22	2.21
	Fruit	473.27	411.76 ± 6.44 c	0.19	8.83	
	Stem	77.91	408.02 ± 10.58 c	0.03	1.44	
	Canes	505.14	441.72 ± 7.88 a	0.22	10.11	
	Perennial branches	1276.38	421.00 ± 5.35 c	0.54	24.35	
	Roots	2268.55	418.72 ± 10.07 c	0.95	43.05	
10-year-old	Leaves	970.31	433.46 ± 6.42 a	0.42	10.67	3.94
	Fruit	1115.31	414.54 ± 12.96 b	0.46	11.73	
	Stem	214.91	416.02 ± 9.43 b	0.09	2.27	
	Canes	1202.47	442.32 ± 6.12 a	0.53	13.49	
	Perennial branches	3019.70	409.84 ± 7.42 b	1.24	31.40	
	Roots	2885.49	415.72 ± 10.75 b	1.20	30.44	
15-year-old	Leaves	1041.61	445.27 ± 5.67 a	0.46	10.49	4.42
	Fruit	2476.01	410.79 ± 11.73 c	1.02	23.00	
	Stem	226.29	408.12 ± 12.11 c	0.09	2.09	
	Canes	1346.32	453.12 ± 2.16 a	0.61	13.79	
	Perennial branches	2754.14	430.27 ± 8.09 b	1.19	26.79	
	Roots	2519.06	418.68 ± 7.05 c	1.05	23.85	
20-year-old	Leaves	1096.99	430.46 ± 6.61 bc	0.47	8.60	5.49
	Fruit	1383.64	422.45 ± 4.01 c	0.58	10.64	
	Stem	245.35	404.00 ± 7.70 d	0.10	1.80	
	Canes	1228.64	455.19 ± 5.98 a	0.56	10.18	
	Perennial branches	3922.72	423.59 ± 8.94 c	1.66	30.25	
	Roots	4829.06	438.26 ± 9.06 b	2.12	38.53	

Notes: Values are the mean ± SD of five biological replicates. Different letters indicate significant differences between samples in multiple comparisons (*p* < 0.05).

**Table 3 plants-12-00997-t003:** The soil carbon storage of each soil layer in the vineyard of different age.

Vine-Age	Soil Layer (cm)	Bulk Density (g/cm^3)^	Carbon Content (g/kg)	Carbon Storage (t·hm^−2^)	Percentage of Total (%)
5-year-old	0–20	1.16 ± 0.05 b	5.81 ± 0.59 a	13.46 ± 1.25 a	28.03
	20–40	1.26 ± 0.05 ab	5.24 ± 0.44 a	12.96 ± 0.94 a	27.00
	40–60	1.35 ± 0.05 a	3.91 ± 0.46 b	10.68 ± 1.01 a	22.24
	60–100	1.34 ± 0.05 ab	2.06 ± 0.58 c	10.91 ± 2.77 a	22.73
10-year-old	0–20	1.19 ± 0.08 b	6.18 ± 0.22 a	14.94 ± 0.80 a	28.30
	20–40	1.33 ± 0.05 a	5.20 ± 0.27 b	13.77 ± 0.86 a	26.09
	40–60	1.33 ± 0.06 a	4.78 ± 0.34 c	12.31 ± 0.91 b	23.33
	60–100	1.35 ± 0.04 a	2.23 ± 0.12 d	11.76 ± 1.06 b	22.28
15-year-old	0–20	1.14 ± 0.07 b	6.73 ± 0.15 a	15.24 ± 0.33 a	27.88
	20–40	1.27 ± 0.06 a	5.36 ± 0.13 b	14.29 ± 0.80 a	26.14
	40–60	1.29 ± 0.04 a	4.82 ± 0.17 c	12.53 ± 0.66 b	22.91
	60–100	1.29 ± 0.02 a	2.45 ± 0.29 d	12.62 ± 1.58 b	23.07
20-year-old	0–20	1.20 ± 0.04 c	6.37 ± 0.50 a	15.80 ± 1.05 a	28.43
	20–40	1.25 ± 0.05 b	5.75 ± 0.16 b	14.38 ± 0.92 b	25.88
	40–60	1.26 ± 0.05 b	4.80 ± 0.22 c	12.31 ± 0.61 c	22.16
	60–100	1.29 ± 0.05 a	2.48 ± 0.16 d	13.08 ± 1.05 c	23.53

Notes: Values are the mean ± SD of five biological replicates. Different letters indicate significant differences between samples in multiple comparisons (*p* < 0.05).

**Table 4 plants-12-00997-t004:** Changes in carbon sequestration in the vineyard ecosystem.

Vine Age Gradient	Vines		Soil		Total Carbon Sequestration (t·ha^−1^·a^−1^)
	Increment of Carbon Storage (t·ha^−1^)	Carbon Sequestration (t·ha^−1^·a^−1^)	Increment of Carbon Storage (t·ha^−1^)	Carbon Sequestration (t·ha^−1^·a^−1^)	
5 year–10 year	1.73	0.35	4.78	0.96	1.31
10 year–15 year	0.48	0.10	1.89	0.38	0.28
15 year–20 year	1.07	0.21	0.89	0.18	0.39

## Data Availability

Data is contained within the article.

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
