# Peer review of "A Simple Method Using an Allometric Model to Quantify the Carbon Sequestration Capacity in Vineyards"

_plants, 2023, doi:10.3390/plants12050997_

Round 1
Reviewer 1 Report
The reviewed manuscript described interesting method of estimation of the carbon sequestration capacity in vineyards. This topic is very important for modern science and agriculture. The abstract reflects the basic ideas of the paper. The Introduction contains the information about the significance of the carbon sequestration. The basic references on the topic are cited in the MS. In the study the appropriate methods were used. The presentation of the results is good. Conclusions are supported by results and discussion. The manuscript is very clear.
I recommend it for publication with minor revision.
Remarks to the authors.
Major comments:
1. Title: What do you think about removing commas from the title of the paper? I think, that “A new simple method using an allometric model to quantify the carbon sequestration capacity in vineyards” looks better.
2. Use Plants template with the line numbering.
Minor comments:
1. Who is the author of Vitis vinifera? (page 2).
2. 2.1. Study site: correct 9.2 ℃ and 10℃ so that the degrees icon does not move to the next line. Correct (figure 1) to (Figure 1).
3. Figure 1: Please, try to make legend designations bigger.
4. Figure 4: Make the labels to the diagrams more visible.
5. Figure 5: Place the diagram in the center of the page.
Reviewer 2 Report
Dear Authors,
I have read with much interest your paper. I think that the topic is of importance for our understanding. However, I found it very difficult to read and understand, therefore it was very difficult to me to judge its value.
Best regards and good luck,
R.
Paper in general: the topic pursued by the authors is of importance in carbon accounting and carbon balance. However, the paper is too poorly written so as to make a fair sense for the reader. I had many difficulties in getting the message in most of its parts. Terminology and language improvements are only small issues which need to be addressed. Format is another thing to be considered. But I think that the most important thing is that of lacking clarity of the methods.
Title: seems to be OK.
Keywords: seem to be OK.
Abstract:
Needs to be double-checked for consistency in the concepts used. Also, abbreviations are discouraged here. For instance, what are 5a, 10a and so on, standing for?
INTRODUCTION
General comment: Citation system seems to be strange. Language needs improvement. The introduction is not too convincing and would need some improvement to more clearly give the gap in knowledge.
Specific comments:
Since there are no line numbers, I am going to refer here to paragraphs.
Paragraph 1: how substitution of non-carbon energy can be a mitigation measure? Check the words used and the language accuracy. What is a cash crop? Isn’t fruit production a cash crop per se? It is uncommon to start a sentence by “such”…
Paragraph 2: second sentence needs rephrasing.
Last paragraph: it is not too convincing as a goal and objectives.
MATERIALS AND METHODS
General comment: it is too poorly written to understand what and why the authors proceed this way.
Specific comments:
Paragraph 1: what do the authors understand by “daily temperature difference”? last sentence: is this the typical way to report the texture? Data shown in this paragraph – any source(s)?
Paragraph 2: not clear for what 5a, 10a, and so on, stand for; vine spacing>>>plant spacing?; plants per ha? Last sentence: predicted from where or by who?
Figure 1: the pictures are nice but they are not related to the map on the left. The legend of the map, on the other hand, needs to be improved for clarity. Maybe the authors could add some points on the map to indicate the location of the sampling.
2.2.
Paragraph 1: first sentence makes no sense. Not clear what the “distinct vines”are. Third sentence: it makes no sense to me how the destructive sampling could be related to the non-wood use. Also, it is not clear to what the authors refer by the sample number reduced as much as possible. Not clear what “standard wood” is. Is the “whole root minim method” something standard? To what are the authors referring by “decomposted”? Have the authors measured the fresh weight? If so, how?
Paragraph 2: is particularly uncarefully written. Many parts make no sense. The phrasing is poor and the use of tense is not OK. Also, it is not clear what, and mostly why, protocols were used for the two drying exercises.
2.3. Determination and calculation method…. ????
Paragraph 1: first sentence: any references to support this statement?
2.3.2. I am not sure if using a tape is a good strategy for diameter measurement…Why only four ranges? Is this statistically sound? Is the randomization useful here to collect the diameters? I would say that it should have been related to the age. Then, it is not clear how the authors have related the mass to the diameter since they have collected a different sample for destructive analysis, then they have measured the base diameters to new plants. These should have been measured to the plants used for mass estimation.
2.3.3. What is soil carbon storage? Also, this section in particularly unclear and difficult to understand. Same in section 2.3.4.
2.4. Seems to be purely an enumeration of steps without any reference to the variables to which these statistical steps were applied and why these tests were applied. Unclear.
RESULTS
General comment: this section is better written in structure but it has shortcomings in format. At least the following should be addressed:
Specific comments:
Title: Why Results and analyses?
First sentence is purely methods. This should have been well explained in the methods. Then, for regression models it is common to report the coefficient of determination and not the coefficient of correlation. All of us would expect the biomass to be correlated to the diameter but what we would like to see is the dependence between them. Last sentence, the same paragraph: grape? Also, here, it is what I said before. Correlation shows the degree of association, not dependence.
Table 1: tables and figures need to be self-explanatory. It is difficult to understand here what are x and y and what are their units of measurement.
3.2. Not clear what the total biomass is here and in the figure below. Then, the approach seems to be wrong because one would like to know how biomass varies by age more than by diameter. That’s because one expects to be more easily to account for age (which may have variation in diameter) that to have equations for diameter within the same age group. Also, in this paragraph the authors report the R2 metric but they refer to correlation.
3.3. Judging by the data in Figure 4 I seriously doubt that there was a normal distribution in left (both) panels and in the bottom right panel. Also, the paragraph says nothing about the metrics of normality check. Only some information which is not readily seen in Figure 4. Figure 4 is unclear, including the names of the axis.
3.4. It says that the carbon storage was calculated but the figure referred here shows the biomass in kg. Figure lacks in clarity and self-explanation: error bars, letters…
3.5.First sentence is something we already know. Therefore it belongs to introduction.
3.6. Is the title accurate? First three sentences have nothing to do with results. Table 3 is not self-explanatory.
DISCUSSION
General comment: in some parts it seems to be OK but most of it is difficult to read and understand.
Specific comments:
Second sentence: it is not clear how. I did not find this in your results. Chiarawipa et al., they...no reference here and in other parts.
CONCLUSION
General comment: how about fertilizers?
Reviewer 3 Report
ï½€
This manuscript uses a simple method to quantify the carbon storage in vineyards based an allometric model. I agree that the carbon storage is an important topic of for plants, including vineyard. Although whole text is easy to understand, some sections should be improved. I provide some comment as follows.
1. For Title, I don’t agree that this is a new method but a simple method. Therefore, title should be improved.
2. I suggested that authors should define “carbon storage” and “carbon sequestration” in introduction because these two terms are not so equal. Therefore, the terms should be defined at the first time in this paper.
3. In abstract and text, I suggested that don’t use such as “15a” because audiences could understand the meaning of that. It should use “15-year-old” to format for whole text.
4. I suggest adding the study purposes in the end of Introduction.
5. Page 2, pointed out “… in the whole year can reach more than 3200 °C”. I think that is not so suitable. Authors might consider using average temperature.
6. Each equation should give number. Please improve all equations.
7. In statistical method, rare studies used Duncan’s in test, especially in scientific papers. Therefore, authors might consider another approaches.
8. Carbon content might consider using %.
Overall, I found that this study is interesting. I recommend this paper for publication in Plants after revised.

Reviewer 4 Report
The manuscript entitled “A new, simple method, using an allometric model to quantify the carbon sequestration capacity in vineyards” submitted for possible publication to “Plants” is an interesting study. The manuscript covers the contents to some extent positively, but there are some key points lacking. The manuscript can be accepted after major revision as per comments given below:
Ø The aims of the study are not elaborated in the abstract, I would suggest to add them.
Ø The methodology is not attractive and describing the key inputs.
Ø Conclusion of the study is missing.
Ø What are the limitations of current study, how the authors would suggest to overcome for future studies.
Ø Remove the keywords that are used in title, and rearrange alphabetically.
Ø What are the major issues generated from climate change.
Ø What are the common causes associated with climate change and how the current agroecosystem is being affected by the climate change operations?
Ø In the first paragraph clearly indicate the issues being addressed in the current study focused on vineyard production.
Ø Update the studies with latest references preferably among 2020-2023
Ø What are the key consequences to perform this study?
Ø The hypothesis tested is missing.
Ø The objectives of the study are lacking.
Ø What is the practical implementation of current study in field agriculture?
Ø Add a sub section in the discussion section to describe the limitations of study in different climatic zones.
Ø Statistical lettering is missing in table 2, also indicate the terms in footnotes of tables and figures.
Round 2
Reviewer 2 Report
Dear Authors,
I have completed my second assessment of this paper. Thank you for considering my comments. This version was significantly improved.
Good luck!
Author Response
Dear reviewer,
We would like to express our great appreciation to you for your second assessment of this paper and thank you very much for your affirmation of our revised content. We rechecked the language and references of this manuscript in round 2. To help readers understand our research better, we simplified some complex sentences without changing their original meaning. The revisions to the manuscript were marked in green.
Thank you again for your attention to our paper. If you have any further queries, please do not hesitate to contact us.
Kind regards,
Dr. Rui Song

Reviewer 4 Report
Now it is ok
Author Response

(The authors gave the same response as above.)
